spectroscopy

resonance Rayleigh scattering, duloxetine, spectrofluorimetric, erythrosine-B, content uniformity

**Author for correspondence:**
Ahmed A. Abu-hassan
e-mail: saif.2016@yahoo.com

# Two facile approaches based on association complex with erythrosine-B for nano-level analysis of duloxetine: application to content uniformity

Sayed M. Derayea[1], Ramadan Ali[2] and
Ahmed A. Abu-hassan[2]

[1]Department of Analytical Chemistry, Faculty of Pharmacy, Minia University, Minia 61519, Egypt
[2]Department of Pharmaceutical Analytical Chemistry, Faculty of Pharmacy, Al-Azhar University, Assiut Branch, Assiut 71524, Egypt

AAA, 0000-0002-5016-1916

Duloxetine is an antidepressant that exhibits its action by preventing the reuptake of serotonin and norepinephrine by neurons. In this analytical study, we developed two facile, sensitive methods for duloxetine analysis. Both methods rely on the formation of binary association complex between erythrosine-B and duloxetine in an acidic medium using spectrofluorimetric and resonance Rayleigh scattering (RRS) techniques. Spectrofluorimetric method simply uses the quenching property of the formed complex on the native fluorescence of erythrosine-B at an emission wavelength of 557.2 nm ($\lambda_{ex} = 528.6$), while RRS is based on detecting the enhancement in the RRS signal at 357.2 nm. The proposed methods have been validated according to the International Conference on Harmonization guidelines. The approaches provide linear assay of duloxetine hydrochloride over 0.1–2.4 µg ml$^{-1}$ and 0.2–2.0 µg ml$^{-1}$ for spectrofluorimetric and RRS methods, respectively. Variables affecting methods and complex formation were studied and optimized. The limit of detection values were 0.03 and 0.056 µg ml$^{-1}$ for spectrofluorimetric and RRS methods, respectively. Both approaches were applied with acceptable results for formulation analysis and evaluation of cymbatex capsule content uniformity.

# 1. Introduction

Duloxetine hydrochloride (DLX) is chemically named (+)-(S)-N-methyl-γ-(1-naphthyloxy)-2-thiophenepropylamine hydrochloride. It is an effective reuptake inhibitor of norepinephrine and serotonin. DLX has no notable affinity for cholinergic, dopaminergic, adrenergic, histaminergic, glutamate, opioid or GABA receptors [1]. It is prescribed for the treatment of major depressive disorder, anxiety disorder management, fibromyalgia, peripheral neuropathic pain in diabetic patients or induced by chemotherapy [2,3], and stress urinary incontinence [4]. DLX has many advantages over other antidepressants such as improved efficacy, safety, tolerance, fewer side effects, dual inhibiting properties and lower affinity for neuronal receptors [5].

The published methods for DLX involve spectrophotometry [6–11], spectrofluorimetric approaches based on native fluorescence property [12,13] or derivatization by NBD-Cl to give fluorescent product measured at 523 nm [14]. Thin-layer chromatography was also reported for the detection of DLX [15,16], as were electrochemical methods [17,18], gas chromatography [19] and high-performance liquid chromatography (HPLC) [20–23].

However, most of the published papers for DLX assay use spectrophotometric techniques that suffer from low sensitivity and comprise an extraction step, which renders the procedure tedious and time-consuming [6–11]. Other techniques used HPLC, which needs ultra-pure solvents, sample pretreatment, expensive detectors and trained personnel.

Erythrosine-B dye is a popular reagent that is used for the analysis of several compounds using spectrophotometry, spectrofluorimetry and resonance Rayleigh scattering (RRS) [24–28]. The used reaction is based on the formation of an association complex with the target compound. The complex is formed due to the electrostatic attraction of the protonated centre of a lipophilic drug and the negatively charged dye, erythrosine-B [29]. The analysis is based on monitoring the fluorescence intensity changes (quenching of erythrosine-B) or improving the intensity of the RRS spectrum. Because DLX has a basic centre (amino group), it can form an ion pair complex with erythrosine-B, permitting its assay by either RRS or spectrofluorimetric approaches. In this study, two facile and sensitive methods were developed for the assay of DLX using erythrosine-B as a fluorescence or Rayleigh scattering probe.

# 2. Experimental

## 2.1. Apparatus

Measurements for both approaches were performed using a spectrofluorometer (Scinco, Korea, serial no. FS-1304002). The instrument was equipped with a Xe-arc lamp (150 W). The slit width of excitation and emission was adjusted at 5 nm, and the photomultiplier detector was set to 400 V. The RRS method was scanned by synchronous spectrofluorimetry ($\lambda_{ex} = \lambda_{em}$; $\Delta\lambda = 0$). pH measurements were performed with a Jenwey type pH meter (EU model 350). An electronic analytical balance with a single pan (Precisa XB 220A, Switzerland) was used for weighing chemicals and raw materials.

## 2.2. Materials, standard solution and reagents

DLX was obtained from the Mash Premiere pharmaceutical company (Badr City, Cairo, Egypt). Cymbalta 30 mg (BN: D314479A) and cymbatex capsules 20 mg (BN: 2005201) were used for dosage form analysis. Erythrosine-B was obtained from Merck (Darmstadt, Germany) and prepared into a 250 ml volumetric flask by dissolving 40 mg into 250 ml water to give a molar concentration of $1.82 \times 10^{-4}$. Teorell–Stenhagen [30] buffer was carefully prepared by mixing suitable volumes of solution A (0.1 M HCl) and solution B (343 ml of 1 M NaOH, 3.5 ml of phosphoric acid and 100 ml of 0.33 M citric acid) in a 1 l volumetric flask filled with water to the mark. The pH required was adjusted using a pH meter. The organic solvents methanol, ethanol, acetone, acetonitrile and dimethylformamide (DMF) were obtained from El Nasr Company (Egypt) and were of analytical grade. Standard DLX solution was simply prepared by dissolving 20 mg of powder in 40 ml of water in a calibrated flask (100 ml) followed by completion by the same solvent to the mark, and then dilutions were made using water to prepare the standard solutions containing different concentrations of DLX.

## 2.3. General assay procedure

The assay steps were carried out in a 10 ml flask, by transferring 1.0 ml from standard solutions of DLX to give 0.1–2.4 µg ml$^{-1}$ for spectrofluorimetry or 0.2–2 µg ml$^{-1}$ for RRS in the measured solution; 1.2 ml

of Teorell–Stenhagen buffer was added to each flask followed by addition of 2 or 1 ml of erythrosine-B for spectrofluorimetry or RRS, respectively. The flask contents were mixed, completed to the mark with water, and the measurements were performed using a spectrofluorometer. In the spectrofluorimetric method, measurements were performed by detecting the quenching value of erythrosine-B native fluorescence by the added DLX at an emission wavelength of 557.2 nm ($\lambda_{ex} = 528.6$). The RRS method used the enhancement of the RRS spectrum at 357.2 nm using synchronous spectrofluorimetry. A blank experiment was performed in both approaches, and all measurements of the corresponding sample against it were corrected.

## 2.4. Procedure for duloxetine hydrochloride assay in dosage forms

For both methods, the steps of the assay were followed to detect DLX in pharmaceutical preparations. Ten cymbalta capsules (30 mg DLX) were evacuated, and the powder was weighed on an electronic balance. A quantity of powder equal to 50 mg of DLX was quantitatively transferred and dissolved in 50 ml of water by sonication for 10 min, then filtered to a calibrated flask (100 ml) and completed to the mark with water. To prepare DLX concentrations within the linear range, a quantity from the flask was serially diluted and the general approach steps were applied to it. For each approach, the DLX content in capsules was calculated from the regression equation using five measurements for each process. As the assay procedure is simple and rapid, both approaches were extended to evaluate DLX content uniformity in cymbatex capsules. Each capsule was individually assayed using the same procedure for the dosage form and the acceptance value was estimated from the respective equation.

# 3. Results and discussion

Xanthene dye as erythrosine-B dye is a popular reagent that is used in spectrophotometry, spectrofluorimetry and RRS [24–28]. The reaction is based on the formation of an association complex with the detected drug while the measurements are based on detecting the fluorescence intensity changes (quenching of erythrosine-B) or improving the strength of the RRS spectrum.

DLX is a basic drug with a $pK_a$ of 9.7; therefore, it will be almost ionized and fully protonated in an acidic medium (pH 3.8). At the same time, erythrosine-B has two dissociation constants ($pK_{a1} = 3.9$ and $pK_{a2} = 5.0$) in aqueous solution [31]. Hence, if the medium is moderately acidic (pH 3.8), only one group in the dye would be dissociated to give the monovalent anion. Although the dye contains two groups that can be ionized (the hydroxyl and carboxyl groups), the hydroxyl group is the most susceptible to ionization owing to the presence of two iodine atoms next to the hydroxyl group. The electron-withdrawing ability of the iodine atoms is very high and effectively can minimize the electron cloud around the oxygen of the hydroxyl group rendering it easier to dissociate even than the carboxylic group [32,33]. Therefore, in a moderately acidic solution (pH 3.8), the hydroxyl group of erythrosine-B molecule will be ionized forming a monovalent anion that carries a negative charge.

The complex is formed due to the electrostatic attraction of protonated centre of the target drug and negatively charged erythrosine-B dye [29] in an acidic medium. In the spectrofluorimetric approach, monitoring depends on measuring the quenching value of the native fluorescence of erythrosine-B at an emission wavelength of 557.2 nm ($\lambda_{ex} = 528.6$) (figure 1). Transformation of the fluorescent erythrosine to a non-fluorescent DLX–erythrosine association complex is the cause of quenching.

## 3.1. Resonance Rayleigh scattering detection

Detection of the RRS spectrum of the product was performed at 357.2 nm using synchronous spectrofluorimetry (figure 2). The RRS enhancement is attributed to molecular volume enlargement, hydrophobic interface formation and rigidity as explained below.

## 3.2. Enlargement of molecular volume

RRS strength is directly influenced by the increase of molecular volume. As estimation of molecular scattering volume is tedious, the ion pair molecular weight can be used as an alternative as described in the formula $I = KCMI_0$, where $I$ represents RRS strength, $K$ is the coefficient constant, $M$ is the molecular weight, $I_0$ is the incident light intensity and $C$ refers to the concentration of the ion pair [27]. When all variables of the equation are constant, the RRS strength is directly proportional to ion

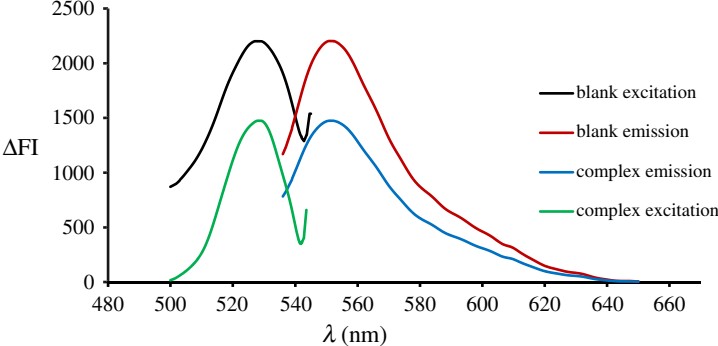

**Figure 1.** Excitation and emission spectra of erythrosine-B ($1.82 \times 10^{-4}$ M) and its association complex with DLX (1 µg ml$^{-1}$).

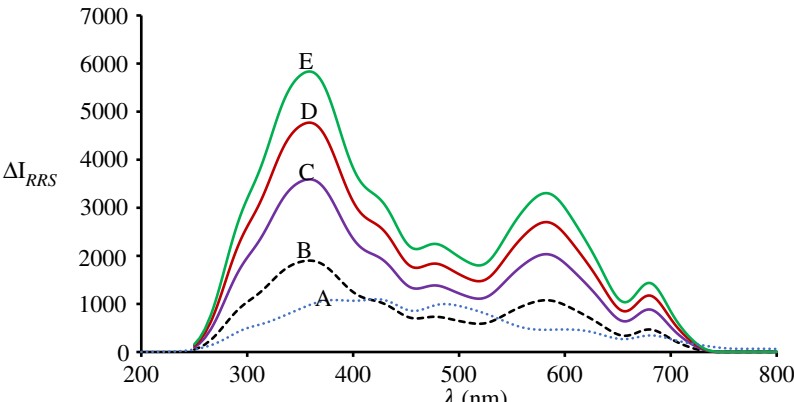

**Figure 2.** RRS spectra at pH 3.8 for DLX (1 µg ml$^{-1}$) (curve A), erythrosine blank (curve B) and erythrosine–DLX complex at three concentrations 0.5, 1.2 and 1.8 µg ml$^{-1}$ (curves C, D and E, respectively).

pair molecular weight. In the presented approach, the increase of ion pair molecular weight was from 289.42 (DLX-H$^{+}$) to 1124.31 (DLX–erythrosine) which in turn enhances RRS strength.

### 3.3. Hydrophobic interface formation

In an acidic medium, erythrosine-B presents as an anion while DLX presents as a protonated cation. Both are capable of the formation of a hydrated ion with water which exhibits a very weak RRS strength [34]. When erythrosine reacts with DLX via association complex, a hydrophobic solid–liquid interface appeared due to aryl framework formation. Hydrophobic interface formation leads to an augmentation of the RRS signal.

### 3.4. Rigidity and molecular planarity effects

After the interaction between DLX and erythrosine via association complex, the aryl group rotation becomes limited and this is attributed to rigidity, molecular planarity strengthening and molecular volume increasing. As a consequence, the enhancement of the scattering intensity was achieved [27].

## 4. Experimental condition study

Variables that could affect the formation of the complex between DLX and erythrosine-B have been carefully studied. Each variable was investigated individually while other variables were held constant. The values of $\Delta$FI and $\Delta I_{RRS}$ were recorded each time, and the optimized variable value was selected.

### 4.1. Buffer pH and volume

The current approaches were carried out at diverse pH (2.8–5) using Teorell buffer. Each time, the difference was reported as fluorescence quenching or enhancement in the RRS value. It was noticeable

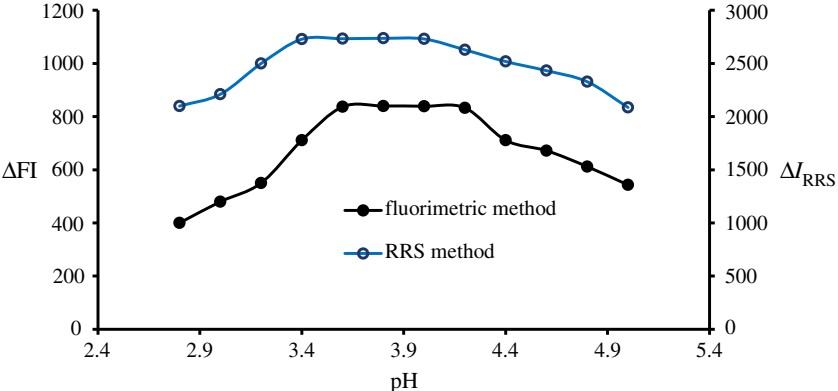

**Figure 3.** Effect of pH on the $\Delta$FI effect and $\Delta I_{RRS}$ of the association complex formed between DLX (1.1 µg ml$^{-1}$) and erythrosine-B.

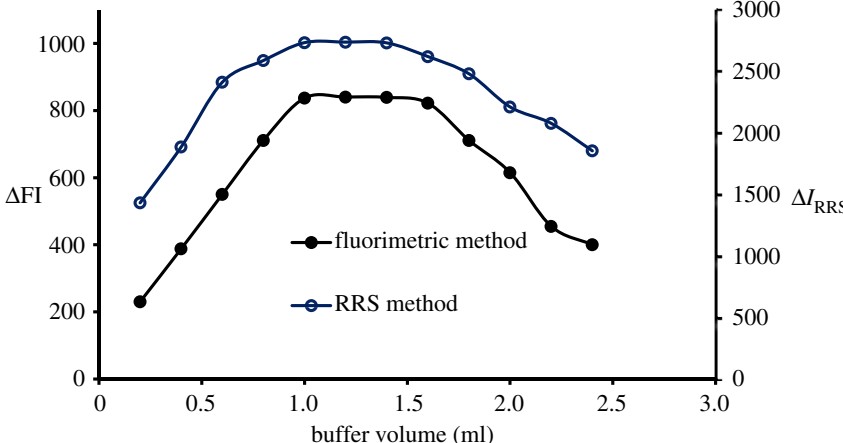

**Figure 4.** Effect of buffer volume on the $\Delta$FI effect and $\Delta I_{RRS}$ of the association complex formed between DLX (1.1 µg ml$^{-1}$) and erythrosine-B.

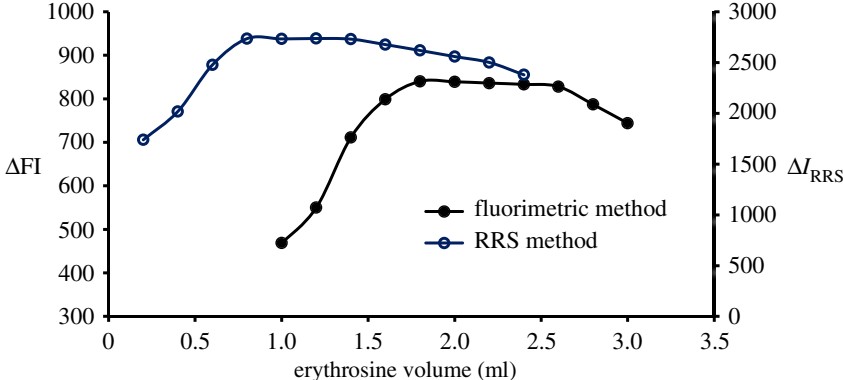

**Figure 5.** Effect of erythrosine-B volume (1.82 × 10$^{-4}$ M) on the $\Delta$FI effect and $\Delta I_{RRS}$ of the association complex formed with DLX (1.1 µg ml$^{-1}$).

that the DLX–erythrosine complex is strongly dependent on the pH value. Higher quenching and $\Delta I_{RRS}$ values were recorded at pH 3.8 (figure 3). Furthermore, the buffer volume was investigated using varied volumes (0.2–2.4), and the optimum volume for both approaches was 1.2 (figure 4).

## 4.2. Erythrosine-B volume and diluting solvent

To choose the optimum volume of erythrosine-B, different volumes of the reagent were added while the quenching or $I_{RRS}$ values were determined. It was found (figure 5) that 2.0 and 1.0 ml of erythrosine-B

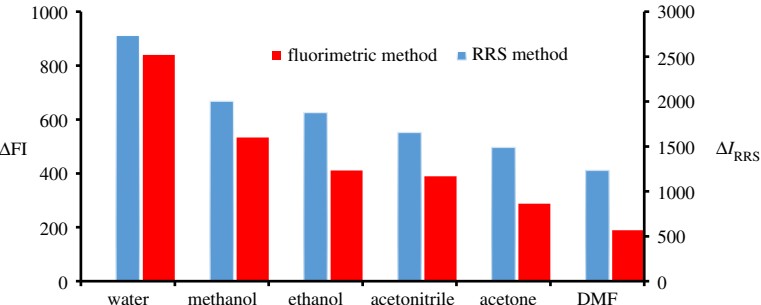

**Figure 6.** Effect of diluting solvent on the $\Delta$FI effect and $\Delta I_{RRS}$ of the association complex formed between DLX (1.1 µg ml$^{-1}$) and erythrosine-B.

**Table 1.** Analytical parameters for the assay of DLX by the suggested spectroscopic methods.

| parameter | fluorimetric method | RRS method |
|---|---|---|
| linear range (µg ml$^{-1}$) | 0.1–2.4 | 0.2–2.0 |
| slope | 570 | 1762.2 |
| standard deviation of slope ($S_b$) | 3.7 | 20.61 |
| intercept | 211.64 | 800.74 |
| standard deviation of intercept ($S_a$) | 5.21 | 29.83 |
| determination coefficient ($r^2$) | 0.9997 | 0.9993 |
| correlation coefficient ($r$) | 0.9998 | 0.9996 |
| number of determinations | 7 | 7 |
| LOQ (µg ml$^{-1}$) | 0.091 | 0.169 |
| LOD (µg ml$^{-1}$) | 0.03 | 0.056 |

are optimal and were selected for fluorimetric and RRS methods. Moreover, different solvents were attempted in the dilution to choose the appropriate one. Water was selected as an appropriate solvent as shown in figure 6 benefiting from availability, low price and compatibility with green chemistry. The stability of the DLX–erythrosine association complex was tested by both approaches at a variety of time intervals and measurements revealed that the complex is stable for 1 h.

## 4.3. Study validation

Criteria of the International Conference on Harmonization (ICH) [35] were followed to validate both methods for quality control applications.

## 4.4. Linearity range and sensitivity

The calibration plot for each methodology was constructed from the data obtained from a triplicate assay of seven DLX concentrations. The graph was constructed by plotting the decrease of erythrosine-B native fluorescence or the increase in the RRS signal value against DLX concentration (µg ml$^{-1}$). Regression equations derived from the graphs and other statistical parameters are summarized in table 1. Linearity appeared at 0.1–2.4 µg ml$^{-1}$ for the spectrofluorimetric method and 0.2–2 µg ml$^{-1}$ for the RRS approach. The sensitivity of the methods was assessed by limit of quantitation (LOQ) and limit of detection (LOD) estimation. Estimates rely on the equations provided by the ICH directives [35]:

$$LOQ = \frac{10\sigma}{\text{slope}} \quad \text{and} \quad LOD = \frac{3.3\sigma}{\text{slope}},$$

where $\sigma$ refers to the standard deviation of the intercept.

The LOD and LOQ were 0.03 and 0.091 µg ml$^{-1}$ for the spectrofluorimetric approach, while for the RRS approach they were 0.056 and 0.169 µg ml$^{-1}$.

**Table 2.** Estimation of accuracy for the proposed spectroscopic methods.

| DLX conc. (µg ml$^{-1}$) | fluorimetric method % recovery ± s.d.[a] | RRS method % recovery ± s.d.[a] |
|---|---|---|
| 0.3 | 98.26 ± 1.47 | 99.04 ± 1.54 |
| 1 | 101.76 ± 0.51 | 100.62 ± 1.14 |
| 1.5 | 98.68 ± 1.23 | 101.35 ± 1.74 |
| 1.8 | 99.48 ± 0.39 | 98.39 ± 0.47 |

[a]Mean of three determinations; s.d., standard deviation.

**Table 3.** Estimation of intra-day precision and inter-day precision for the suggested spectroscopic methods.

| method | DLX conc. (µg ml$^{-1}$) | inter-day precision % recovery ± RSD[a] | intra-day precision % recovery ± RSD[a] |
|---|---|---|---|
| fluorimetric | 0.5 | 101.65 ± 1.59 | 98.25 ± 1.69 |
| | 1.1 | 99.69 ± 0.51 | 101.07 ± 0.75 |
| | 1.9 | 99.51 ± 0.84 | 98.89 ± 1.32 |
| RRS | 0.5 | 100.41 ± 0.91 | 99.58 ± 1.10 |
| | 1.1 | 98.84 ± 0.65 | 101.13 ± 0.86 |
| | 1.9 | 98.98 ± 0.89 | 101.04 ± 1.81 |

[a]Mean of three determinations; RSD, relative standard deviation.

## 4.5. Accuracy and precision

To test the accuracy of the approaches, four concentrations of DLX standard solution within linear range were analysed, and triplicate calculations of each were recorded. The calculations are listed as % recovery ± s.d. in table 2 and revealed acceptable agreements between measured and true results. The precision of the two approaches was studied at low and intermediate levels using three DLX standard concentrations. Intra-day precision was evaluated by analysis on the same day while inter-day was evaluated by performing assay on three consecutive days. The lower relative standard deviation value gives evidence for the precision (table 3).

## 4.6. Robustness

The robustness of the developed spectrofluorimetric and RRS methodologies was examined by performing the assay with a slight intended variation of the actual value of buffer volume, pH and erythrosine volume. The results showed no significant effect due to a minor variance of the conditions (table 4).

# 5. Application

## 5.1. Dosage form application

The current spectroscopic approaches were applied for DLX analysis in cymbalta capsules. The estimated percentage values of recovery were 98.02 ± 0.95 and 101.59 ± 1.86 for the spectrofluorimetric and RRS approaches. Additionally, the same product was analysed using a previously published method [14], and the proposed methods were statistically compared via F-test and Student's t-test. No significant changes were observed between proposed and published methods (table 5) as values of the F-test and Student's t-test were smaller than tabulated at a confidence limit of 95%. A comparison of the current work with some reported methods [36–39] is outlined in table 6.

## 5.2. Content uniformity testing

Testing the drug uniformity within capsules is recommended when the drug content in the capsules is less than 25 mg or its percentage is less than 25% of the capsule components [40,41]. However, testing the

**Table 4.** Robustness study of the proposed methods for determination of DLX (1.1 µg ml$^{-1}$) in pure form.

| | % recovery ± s.d.[a] | |
|---|---|---|
| parameter | fluorimetric method | RRS method |
| pH of solution | | |
| 3.6 | 98.99 ± 0.80 | 98.96 ± 0.77 |
| 4 | 101.28 ± 1.04 | 101.32 ± 1.18 |
| volume of erythrosine-B (ml) | | |
| optimum + 0.2 | 98.57 ± 1.36 | 98.91 ± 0.76 |
| optimum − 0.2 | 100.48 ± 0.75 | 101.63 ± 1.24 |
| volume of buffer (ml) | | |
| 1.4 | 99.79 ± 0.60 | 98.98 ± 0.85 |
| 1.0 | 101.28 ± 1.04 | 101.34 ± 1.21 |

[a]Mean of three replicate measurements; s.d., standard deviation.

**Table 5.** Analysis of DLX in dosage form by reported [14] and proposed methods.

| | % recovery ± s.d.[a] | reported method % recovery ± s.d.[a] | t-test value[b] | F-test value[b] |
|---|---|---|---|---|
| fluorimetric method | 98.02 ± 0.95 | 98.25 ± 1.63 | 0.28 | 2.95 |
| RRS method | 101.59 ± 1.86 | 99.68 ± 0.79 | 2.12 | 5.50 |

[a]Average of five determinations.
[b]Tabulated value at 95% confidence limit; $F = 6.338$ and $t = 2.306$.

content uniformity in each capsule is strenuous and lengthy. Therefore, the current work has the merits of a simple and rapid assay as the complex formed instantaneously does not require any extraction or heating which is time-consuming. These merits allow direct investigation of cymbatex capsules by both approaches. As known from directives of the United States pharmacopeia, the acceptance value can be estimated from the following formula: $AV = |M - \bar{X}| + KS$, where $M$ is a reference value, $K$ is the acceptability constant (equal to 2.4 in the case of 10 capsules), $S$ is the sample standard deviation and $\overline{X}$ is the mean of the % recovery of individual content. The AV should be lower than the maximum allowed AV (L1 = 15). The above equation will be modified according to the value of $\bar{X}$.

— If $98.5\% \le \bar{X} \le 101.5\%$, then $M = \bar{X}$ (AV $= KS$).
— If $\bar{X} < 98.5\%$, then $M = 98.5\%$ (AV $= 98.5 - \bar{X} + KS$).
— If $\bar{X} > 101.5\%$, then $M = 101.5\%$ (AV $= \bar{X} - 101.5 + KS$).

The data shown in table 7 revealed the uniformity of cymbatex capsules because the acceptance value was less than the maximum allowed value.

## 5.3. Molar ratio determination

The molar ratio between DLX and erythrosine-B was investigated by Job's method [42]. Master solutions of DLX and dye were prepared in equimolar concentration of $9.09 \times 10^{-4}$. Into a 10 ml flask, a complementary volume equal to 1 ml of both DLX and dye was added in different mole factions (0.1–0.9). The procedure was completed as in the general assay of RRS and spectrofluorimetric methods. All values were corrected against blank. Job's plot results revealed that the molar ratio between DLX and dye is 1 : 1 as only one basic moiety is present in DLX (figure 7). The DLX–erythrosine association complex is elucidated in scheme 1.

**Table 6.** Comparison of the proposed methods with some reported methods.

| method | principle of assay | linear range (µg ml$^{-1}$) | LOQ (µg ml$^{-1}$) | LOD (µg ml$^{-1}$) | heating or extraction step | application |
|---|---|---|---|---|---|---|
| proposed spectrofluorimetric method | spectrofluorimetric determination of the ion pair complex formation with erythrosine | 0.1–2.4 | 0.091 | 0.03 | absent | dosage form and content uniformity testing |
| proposed RRS method | spectrofluorimetric determination of the ion pair complex formation with erythrosine | 0.2–2.0 | 0.169 | 0.056 | absent | dosage form and content uniformity testing |
| reported method [36] | spectrofluorimetric determination of the native fluorescence in 0.05 M acetic acid. | 0.020–0.400 | 0.01 | 0.003 | absent | capsule dosage forms |
| reported method [37] | spectrofluorimetric determination of the native fluorescence in acidic medium | 0.3–30 | 1.69 | 0.56 | absent | stability study and capsule dosage form |
| reported method [14] | spectrofluorimetric after derivatization with NBD-Cl | 0.05–0.25 | 0.003 | 0.001 | present | capsule dosage form and spiked plasma |
| reported method [38] | micellar enhanced native fluorescence | 0.001–0.07 | 0.001 | 0.0005 | absent | capsule dosage form |
| reported method [39] | HPLC after pre-column derivatization with NBD-Cl and fluorescence detection | 0.01–0.6 | 0.001 | 0.0005 | present | capsule dosage forms |

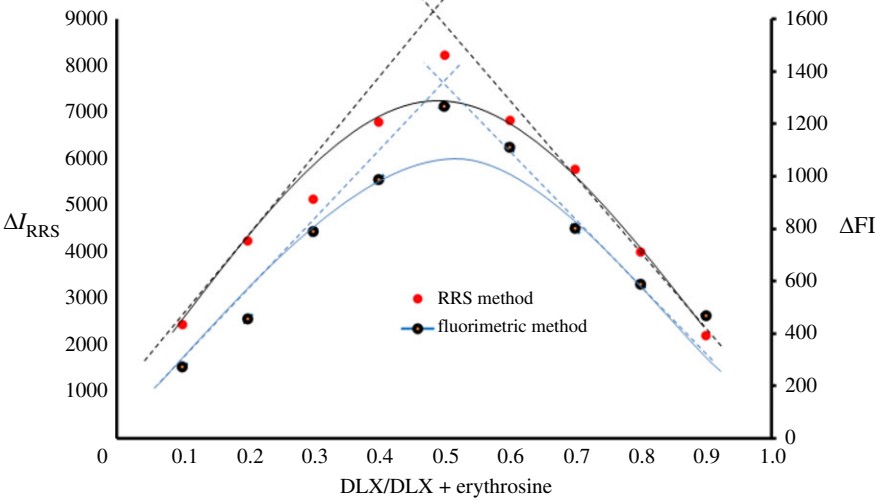

**Figure 7.** Job's plot for molar ratio determination between DLX and erythrosine using equimolar concentration.

**Scheme 1.** The proposed pathway for association complex formation between DLX and erythrosine.

**Table 7.** Content uniformity evaluation for cymbatex capsules (20 mg tablets) by the suggested spectroscopic methods.

| parameter | fluorimetric method | RRS method |
|---|---|---|
| recovery | 97.67 | 97.78 |
| | 101.28 | 102.56 |
| | 96.81 | 103.39 |
| | 99.26 | 102.85 |
| | 97.56 | 101.15 |
| | 102.56 | 102.22 |
| | 97.24 | 96.17 |
| | 101.28 | 103.03 |
| | 97.13 | 103.04 |
| | 97.08 | 102.96 |
| mean ($\bar{X}$) | 98.79 | 101.51 |
| s.d. | 2.15 | 2.50 |
| acceptance value | 5.16 | 6.01 |
| maximum allowed acceptance value | | 15 |

# 6. Conclusion

A spectroscopic study for the feasible, rapid and sensitive analysis of DLX via spectrofluorimetry and RRS is reported. The study focused on the formation of an association complex between DLX and erythrosine-B. In an acidic medium, the association complex is simply formed as a result of electrostatic attraction of opposite charges of both DLX and erythrosine-B. Spectrofluorimetric measurements are based on detecting the quenching of the native fluorescence of erythrosine-B due to the addition of DLX, while the RRS approach uses the enhancement of the RRS spectrum of erythrosine at 357.2 nm. Criteria of the ICH were followed to validate both methods. The current work has preference over some previously reported methods, with acceptable sensitivity, ease of sample preparation, minimized glassware usage and rapidity and does not require extraction or heating steps. Furthermore, the procedures were used to determine DLX in its dosage form and for evaluating content uniformity with capsules.

Data accessibility. This article does not contain any additional data.
Authors' contributions. A.A.A. carried out the experimental work, participated in data analysis and drafted the manuscript. R.A. carried out the statistical analyses and helped draft the manuscript. S.M.D. designed and coordinated the study, and revised the manuscript critically in its final form. All authors gave final approval for publication.
Competing interests. We declare we have no competing interests.
Funding. We received no funding for this study.

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
