## [Peer Review File · Royal Society Open Science]

Review History

RSOS-210922.R0 (Original submission)

Review form: Reviewer 1 (Walaa Abd-ALGhafar)

Is the manuscript scientifically sound in its present form?

Yes

Are the interpretations and conclusions justified by the results?

Yes

Is the language acceptable?

Yes

Do you have any ethical concerns with this paper?

No

Have you any concerns about statistical analyses in this paper?

No

Recommendation?

Accept with minor revision (please list in comments)

Comments to the Author(s)

This is an interesting study and the authors have developed two simple methods and applicable in quality control labs with wide linearity range. The paper is generally well written and structured. The authors discussed the study well and give reasons for the results.

The author performed the optimization of variables, and validated the proposed method with previously reported method, with respect to accuracy and precision, and I recommend publication in Royal Society Open Science with minor modifications:

- Who is the corresponding author? Please refer to him in affiliation.
- The word (xantheme) in the paragraph under Results and discussion should be (Xantheme) first letter is capital.
- There should be 10 spaces at the beginning of the paragraphs in all manuscript.

Review form: Reviewer 2**Is the manuscript scientifically sound in its present form?**

No

Are the interpretations and conclusions justified by the results?

Yes

Is the language acceptable?

No

Do you have any ethical concerns with this paper?

No

Have you any concerns about statistical analyses in this paper?

Yes

Recommendation?

Major revision is needed (please make suggestions in comments)

Comments to the Author(s)

In the present study, the authors developed TWO quantitative approaches to determine the duloxetine, one is spectrofluorometry and the other is resonance Rayleigh scattering..

- In the part of the Abstract:
 - o Line 29: Kindly add the abbreviation of Resonance Rayleigh Scattering, RRS.
 - o In the abstract, line 34, the linearity was 100 – 2400 ng/mL, it is better to consider the number significance writing to be replaced by 0.1 – 2.4 μ g/mL. In the same context, the linearities are nearly the same as 0.1 or 0.2 to 2.4 or 2.0 μ g/mL, for both methods spectrofluorimetry and RRS, respectively. BUT it was expected that if you reached the ideal conditions of the resonance light-scattering (RLS) technique, the optical properties of the resonance light scattering has a magnitude higher than light emission, could you explain that WHY??
 - o The LOD of the fluorimetric method is 0.030 μ g/mL, BUT for RRS was 0.056 μ g/mL, that the fluorimetric method is more sensitive, why extend the other procedure, RRS, give the resonance.
- In the part of the Introduction

- o In line 19: replace the word “methodology” with “method”, kindly, as you talked about the surveys, interviews, experiments, and other investigatory processes, etc.
- In the part of Materials, standard solution, and reagents_line 27
- o The mention names of the dosage forms, Cymbalta and Cymbatex, it is advised to remove the trad name to guard your work against any conflict of interest, related to these companies.
- o In witting of the standard solution, you have ONLY to write the final concentrations of stock solutions, and working solution, or quality control sample as well. Kindly remove the unwanted details.
- In the part of general assay procedure:
 - o The used indicative concentration units are different, for drug ng/mL, and for the reagent was molarity, it is better to unify these units.
- In the the Results and discussion part:
 - o Line 39, use the name erythrosine-B in the whole manuscript.
- In the part of Results and Discussion
 - o In line 9: you did not mention the fixed value of $\Delta \lambda$, nm, is used in synchronous activity??
 - In the part of Mechanism of augmentation of resonance Raleigh scattering
 - o In the equation of
 - = o where I represent RRS strength, K coefficient constant, M M.Wt, I the incident intensity, C refer to solution concentration.
 - HOW you calculated all these variables??? You have to clarify more discussion.
 - M.Wt: is this an official abbreviation??
 - o In line 38: “when Erythrosine reacts with DLX via association complex, a hydrophobic solid-liquid interface appeared leads to an augmentation of the RRS signal”?? clarified the formed complex in the form of nanoparticles of discussion that is required for RRS?? Mention more discussion??
 - o In line 48: “consequence, the enhancement of the scattering intensity was achieved”. In the reported reference you cited, they studied the interaction of erythrocin-B with R- and S-propranolol; the drug was used is different than DLX, did you confirm the rigidity of the formed complex by another way,?? Clarify??
 - In the part of Experimental conditions study:
 - o In the part of the Buffre pH and its volume:
 - In line 15: the written pH value was “pH 3.8 ± 0.2 ”, I think NO mean to calculate the pH average?? Kindly check the value again and ONLY write one number such as “3.8”.
 - In line 17: the authors mentioned that “Furthermore, the buffer volume was investigated at varied volumes (0.2 - 2.4) and the optimum volume for both approaches was 1.2 ± 0.2 . I think as well no need to write the average of the used volume and should be one value, with its metric uint, such as mL. kindly check again.
 - o In the part of Linearity range and sensitivity:
 - In the sentence “The calibration plot for each methodology”, there is a difference between method and methodology, kindly check and select the proper term here.
- o In the part Accuracy and precision
 - The authors mentioned that “our concentrations of DLX within linearity range were analyzed and triplicate estimation of each one was recorded. The calculations were listed as % recovery \pm SD”. Is this study was applied for the standard drug or the extracted sample?? Clarify.
 - In the study of accuracy and precision; the authors used different quality control concentrations, WHY??
 - In table 2 and 3: noted that the value of the higher concentrations was low precise than the low values of concentration. Clarify??
- o In the part of Applictaion

- The authors mentioned the trade name of the used dosage form of the present drug “Cymbalta”. It is better to blind the name of the marketed dosage forms, to avoid the conflict of interest.
- In the sentence “The estimated percentage values of recovery were 98.02 ± 0.95 and 101.59 ± 1.86 for fluorimetric and RRS approaches”. Respectively, 1 mg/capsules (conc/capsules 30 mg) is lost and added, after applied both methods, it is better to mentioned the batch number and manufacture date accordingly.
- In the part of the “content uniformity test”
 - The authors mentioned that “As known from directives of USP, the acceptance value estimated from the following formula $= | \frac{M}{S} - 1 | \times 100$ where; M is a reference value, \bar{x} is mean of recoveries of label claim, S is the SD of recoveries, and K is the constant of acceptability.
 - o Kindly put more details and references.
 - o What is the value of K is the constant of acceptability, is used during the study.
 - In Table 4: there are values of recoveries (96.17, 96.81, 97.08, ...etc) less than the recorded in the part of the application, using the same cymbatex tablets concentration of, 20 mg/tablet. Clarify the shifted accuracy to these values.
 - In line 22 “cympatex capsules 20 mg tablets”, is correct??
 - The authors mentioned in the part of methods and materials two kinds of the marketed drug “Cymbalta and Cymbatex”, but the content uniformity ONLY applied for one (Cymbatex tablets), could give the reasons.
 - Finally, it is better to add further discussion to clarify the obtained results. Other miswriting and suggestion are present throughout the body of the pdf, kindly refer and reply.

Review form: Reviewer 3

Is the manuscript scientifically sound in its present form?

Yes

Are the interpretations and conclusions justified by the results?

Yes

Is the language acceptable?

Yes

Do you have any ethical concerns with this paper?

No

Have you any concerns about statistical analyses in this paper?

No

Recommendation?

Major revision is needed (please make suggestions in comments)

Comments to the Author(s)

The manuscript can be accepted after suitable revision. The revised manuscript should address the following comments:

1- Language should be carefully revised throughout the manuscript.

2- Some related articles could be found in the scientific literature. Authors should check these articles, compare their method to those published and cite whatever relevant to their research. Please find them below.

3- Table 1: The intercept values are so large relative to the slope values. I wonder what would be the reason for such high intercept values. Please correct or explain.

Darvishi, E., Shekarbeygi, Z., Yousefinezhad, S., Izadi, Z., Saboury, A.A., Derakhshankhah, H., Varnamkhasti, B.S.

Green synthesis of nanocarbon dots using hydrothermal carbonization of lysine amino acid and its application in detection of duloxetine

(2021) Journal of the Iranian Chemical Society, .

ABSTRACT: Depression is a mood disorder in which a person feels tired and bored and also unwilling to do daily activities. Duloxetine is a drug that is used to the treatment of depression and anxiety. Due to the use of different medications to treat the depression and its possible side effects, quick and accurate identification of these drugs is necessary. Also, because of the possibility of suicide in depressed people, rapid detection of drug type in drug poisoning (drug overdose) is crucial. Therefore, various sensors are used, that the most straightforward, and most accessible sensors are optical types. One of the best, simplest and safest fluorescent sensors were used for optical sensors is nanocarbon dots. In this study, a new, inexpensive and green optical biosensor was designed, and fabricated using lysine-based carbon dots to detect detection of Duloxetine. Fluorescent carbon dot was prepared by hydrothermal method. The green carbon dots were characterized by UV-visible spectroscopy, TEM, XRD and zeta sizer. Also, fluorescence of carbon dot was investigated. The CDs are spherical and the average size of the monodisperse nanoparticles was around 15 nm. The X-ray diffraction pattern represents a weak crystalline property that confirms the amorphous phase of carbon dots. The value of quantum yield for carbon dots was 31.3% to standard Quinone sulfate. The detection limit of Duloxetine was 0.002 μM . The recovery of Duloxetine was 99.2 to 101.5%, which indicates this nanosensor has a good ability to detect Duloxetine at low concentrations. The results indicate L-lysine-based CDs can be used professionally and selectively to detect of Duloxetine in real samples and human blood plasma. © 2021, Iranian Chemical Society.

Chadha, R., Bali, A.

Stability indicating spectrofluorimetric method for determination of duloxetine hydrochloride in bulk and in dosage form

(2015) Der Pharmacia Lettre, 7 (7), pp. 232-240.

ABSTRACT: Duloxetine (DLX), is a selective serotonin-norepinephrine reuptake inhibitor (SNRI) recommended for maintenance treatment of major depressive disorder, neuropathic pain especially diabetic polyneuropathy (first-line treatment), generalized anxiety disorder, stress urinary incontinence and fibromyalgia. The present investigation describes the validation of rapid, sensitive, cost effective and reproducible stability indicating spectrofluorometric methods based on the native fluorescence of duloxetine HCl in acidic medium for the estimation of duloxetine HCl in bulk and in formulations. The fluorescence intensity of duloxetine hydrochloride was measured at 336 nm after excitation at 290 nm. The methods were validated with respect to linearity, accuracy, precision and robustness. Linearity was observed in the concentration range of 0.3-30 $\mu\text{g/ml}$ with an excellent correlation coefficients (r^2) ranging from 0.9940-0.9996. The limits of assay detection values were found to range from 0.56-0.89 $\mu\text{g/ml}$ and quantitation limits ranged from 1.69-2.42 $\mu\text{g/ml}$ for the proposed methods. The proposed method was applicable to the determination of the drug in capsules and the percentage recovery was found to range from 99.53 \pm 99.66%. The proposed methods were developed as stability indicating procedures by carrying out the analysis for duloxetine hydrochloride on stressed samples prepared under various forced degradation conditions.

Sagirli, O., Toker, S.E., Önal, A.

Development of sensitive spectrofluorimetric and spectrophotometric methods for the determination of duloxetine in capsule and spiked human plasma

(2014) *Luminescence*, 29 (8), pp. 1014-1018.

ABSTRACT: A new, sensitive and selective spectrofluorimetric method has been developed for the determination of duloxetine (DLX) in capsule and spiked human plasma. DLX, as a secondary amine compound, reacts with 7-chloro-4-nitrobenzofurazone (NBD-Cl), a highly sensitive fluorogenic and chromogenic reagent used in many investigations. The method is based on the reaction between the drug and NBD-Cl in borate buffer at pH 8.5 to yield a highly fluorescent derivative that is measured at 523 nm after excitation at 478 nm. The fluorescence intensity was directly proportional to the concentration over the range 50-250 ng/mL. The reaction product was also measured spectrophotometrically. The relation between the absorbance at 478 nm and the concentration is rectilinear over the range 1.0-12.0 μ g/mL. The methods were successfully applied for the determination of this drug in pharmaceutical dosage form. The spectrofluorimetric method was also successfully applied to the determination of duloxetine in spiked human plasma. The suggested procedures could be used for the determination of DLX in pure form, capsules and human plasma being sensitive, simple and selective. Copyright © 2014 John Wiley & Sons, Ltd.

Alarfaj, N.A., Ammar, R.A., El-Tohamy, M.F.

Cationic-enhanced spectrofluorimetric method for determination of selective serotonin reuptake inhibitor duloxetine hydrochloride in its dosage forms

(2013) *Asian Journal of Chemistry*, 25 (11), pp. 6416-6420.

ABSTRACT: A highly sensitive, rapid, accurate and precise spectrofluorimetric method was developed for the determination of duloxetine hydrochloride in its pharmaceutical formulations. The proposed method is based on investigation of the fluorescence spectral behaviour of duloxetine hydrochloride in cetyl trimethylammonium bromide (CTAB) micellar system. In aqueous solution of borate buffer pH 9.9, the fluorescence intensity of duloxetine hydrochloride was greatly enhanced, 3-fold enhancement, in the presence of cetyl trimethylammonium bromide. The fluorescence intensity of duloxetine hydrochloride was measured at 382 nm after excitation at 275 nm. The fluorescence-concentration plot was rectilinear over the range of 1-70 ng/mL with lower detection limit of 0.5 ng/mL. The method was successfully applied to the analysis of duloxetine hydrochloride in its commercial dosage forms. The results were in good agreement with those obtained with the reported method. The application of the proposed method was extended to the stability studies of duloxetine hydrochloride after exposure to different forced degradation conditions, such as acidic, alkaline, oxidative and thermal conditions, according to ICH guidelines.

Misiuk, W.

Spectrofluorimetric study on inclusion interaction of β -cyclodextrin with duloxetine and its analytical application

(2012) *Indian Journal of Chemistry - Section A Inorganic, Physical, Theoretical and Analytical Chemistry*, 51 (12), pp. 1706-1710.

ABSTRACT: Fluorescence study on inclusion interaction of duloxetine in β -cyclodextrin shows significant increase in the fluorescence of duloxetine in the presence of β -cyclodextrin. The effects of pH and cyclodextrin concentration on the fluorescence spectra are reported. Fluorescence spectroscopy of the host-guest interaction between duloxetine and β -cyclodextrin shows formation of inclusion complex with a 1:1 stoichiometric ratio. The changes in fluorescence of duloxetine on inclusion in the hydrophobic β -cyclodextrin cavity is used to calculate its association constants by non-linear regression method. A new spectrofluorimetric method is proposed for quantitative determination of duloxetine in dosage forms with limit of detection of 2.0×10^{-8} mol/L. The linear range is 5.16×10^{-8} mol/L to 1×10^{-5} mol/L and correlation coefficient is 0.9998. The accuracy (recovery 99.5-100.8 %) and precision (RSD 0.58-0.89 %) values of the proposed method are satisfactory.

Ulu, S.T.

Determination and validation of duloxetine hydrochloride in capsules by HPLC with pre-column derivatization and fluorescence detection

(2012) *Journal of Chromatographic Science*, 50 (6), pp. 494-498.

ABSTRACT: A high-performance liquid chromatographic (HPLC) method is described for the determination of duloxetine hydrochloride in capsules. The method was based on pre-column derivatization with 4-chloro-7-nitrobenzo-2-oxa-1,3-diazole using the fluorimetric detection technique. Duloxetine hydrochloride was analyzed by HPLC using an Inertsil C18 column (5 μ m, 150 \times 4.6 mm) and mobile phase consisted of methanol and water (65:35, v/v). The fluorescence detector was adjusted at excitation and emission wavelengths of 461 and 521 nm, respectively. The linearity of the method was in the range of 10-600 ng/mL. Limits of detection and quantification were 0.51 and 1.53 ng/mL, respectively. The proposed method was successfully applied for determination of duloxetine hydrochloride in its pharmaceutical preparation. The results were in good agreement with those obtained using a reference method. © The Author [2012].

Liu, X., Du, Y., Wu, X.

Study on fluorescence characteristics of duloxetine hydrochloride

(2008) *Spectrochimica Acta - Part A: Molecular and Biomolecular Spectroscopy*, 71 (3), pp. 915-920.

ABSTRACT: The fluorescence characteristics of duloxetine hydrochloride are studied in this paper. The fluorescence emission spectra of duloxetine demonstrate that intramolecular charge-transfer takes place between thiophene ring and naphthalenyloxy group upon irradiation. The effects of excitation light, solvent system, variation of solution pH value, metal ions and vitamin C on the fluorescence spectra of duloxetine hydrochloride are elucidated, respectively. A spectrofluorometric method of quantitative determination of duloxetine in dosage form is reported for the first time, the linear range is 7.14×10^{-8} mol/L to 1.43×10^{-5} mol/L, the linear correlation coefficient r is equal to 0.9997, and the detection limit is 3.5×10^{-8} mol/L. The accuracy and the precision are satisfactory. © 2008 Elsevier B.V. All rights reserved.

Prabhu, S., Shahnawaz, S., Kumar, C., Shirwaikar, A.

Spectrofluorimetric method for determination of duloxetine hydrochloride in bulk and pharmaceutical dosage forms

(2008) *Indian Journal of Pharmaceutical Sciences*, 70 (4), pp. 502-503.

ABSTRACT: A simple accurate, sensitive and reproducible spectrofluorimetric method was developed for the analysis of duloxetine hydrochloride in pure and pharmaceutical dosage form. Duloxetine hydrochloride showed strong native fluorescence in 0.05 M acetic acid having excitation at 225 nm and emission at 340 nm. Effect of different solvents were thoroughly investigated. The calibration graph was linear in the range from 0.020 to 0.400 μ g/ml. The proposed method was statistically validated and successfully applied for analysis of capsule dosage forms. The limit of detection and limit of quantification were found to be 0.003 μ g/ml and 0.010 μ g/ml, respectively. The percentage recovery was found to be in the range of 98.71% to 99.17%.

Decision letter (RSOS-210922.R0)

Dear Mr Abu-hassan:

Title: Two facile approaches based on association complex with erythrosine-B for nano-level analysis of duloxetine. Application to content uniformity
Manuscript ID: RSOS-210922

The editor assigned to your manuscript has now received comments from reviewers. We would like you to revise your paper in accordance with the referee and Subject Editor suggestions which can be found below (not including confidential reports to the Editor). Please note this decision does not guarantee eventual acceptance.

Please submit your revised paper before 21-Aug-2021. Please note that the revision deadline will expire at 00.00am on this date. If we do not hear from you within this time then it will be assumed that the paper has been withdrawn. In exceptional circumstances, extensions may be possible if agreed with the Editorial Office in advance. We do not allow multiple rounds of revision so we urge you to make every effort to fully address all of the comments at this stage. If deemed necessary by the Editors, your manuscript will be sent back to one or more of the original reviewers for assessment. If the original reviewers are not available we may invite new reviewers.

RSC Associate Editor:
Comments to the Author:
(There are no comments.)

RSC Associate Editor:
Comments to the Author:
(There are no comments.)

RSC Subject Editor:
Comments to the Author:
(There are no comments.)

Reviewers' Comments to Author:

Reviewer: 1

Comments to the Author(s)

This is an interesting study and the authors have developed two simple methods and applicable in quality control labs with wide linearity range. The paper is generally well written and structured. The authors discussed the study well and give reasons for the results.

The author performed the optimization of variables, and validated the proposed method with previously reported method, with respect to accuracy and precision, and I recommend publication in Royal Society Open Science with minor modifications:

- Who is the corresponding author? Please refer to him in affiliation.
- The word (xanthene) in the paragraph under Results and discussion should be (Xanthene) first letter is capital.
- There should be 10 spaces at the beginning of the paragraphs in all manuscript.

Reviewer: 2

Comments to the Author(s)

In the present study, the authors developed TWO quantitative approaches to determine the duloxetine, one is spectrofluorometry and the other is resonance Rayleigh scattering..

- In the part of the Abstract:

o Line 29: Kindly add the abbreviation of Resonance Rayleigh Scattering, RRS.

o In the abstract, line 34, the linearity was 100 - 2400 ng/mL, it is better to consider the number significance writing to be replaced by 0.1 - 2.4 μ g/mL. In the same context, the linearities are nearly the same as 0.1 or 0.2 to 2.4 or 2.0 μ g/mL, for both methods spectrofluorimetry and RRS, respectively. BUT it was expected that if you reached the ideal conditions of the resonance light-scattering (RLS) technique, the optical properties of the resonance light scattering has a magnitude higher than light emission, could you explain that WHY??

o The LOD of the fluorimetric method is 0.030 μ g/mL, BUT for RRS was 0.056 μ g/mL, that the fluorimetric method is more sensitive, why extend the other procedure, RRS, give the resonance.

- In the part of the Introduction

o In line 19: replace the word "methodology" with "method", kindly, as you talked about the surveys, interviews, experiments, and other investigatory processes, etc.

- In the part of Materials, standard solution, and reagents_line 27

o The mention names of the dosage forms, Cymbalta and Cymbatex, it is advised to remove the trad name to guard your work against any conflict of interest, related to these companies.

o In witting of the standard solution, you have ONLY to write the final concentrations of stock solutions, and working solution, or quality control sample as well. Kindly remove the un-wanted details.

- In the part of general assay procedure:

o The used indicative concentration units are different, for drug ng/mL, and for the reagent was molarity, it is better to unify these units.

- In the the Results and discussion part:

o Line 39, use the name erythrosine-B in the whole manuscript.

- In the part of Results and Discussion

o In line 9: you did not mention the fixed value of $\Delta \lambda$, nm, is used in synchronous activity??

- In the part of Mechanism of augmentation of resonance Raleigh scattering

o In the equation of

$I_{sc} = K I_0 C^2$ where I represent RRS strength, K coefficient constant, M M.Wt, I the incident intensity, C refer to solution concentration.

□ HOW you calculated all these variables??? You have to clarify more discussion.

□ M.Wt: is this an official abbreviation??

o In line 38: "when Erythrosine reacts with DLX via association complex, a hydrophobic solid-liquid interface appeared leads to an augmentation of the RRS signal"?? clarified the formed complex in the form of nanoparticles of discussion that is required for RRS?? Mention more discussion??

o In line 48: "consequence, the enhancement of the scattering intensity was achieved". In the reported reference you cited, they studied the interaction of erythrocin-B with R- and S-propranolol; the drug was used is different than DLX, did you confirm the rigidity of the formed complex by another way?? Clarify??

- In the part of Experimental conditions study:

o In the part of the Buffre pH and its volume:

□ In line 15: the written pH value was "pH 3.8 ± 0.2", I think NO mean to calculate the pH average?? Kindly check the value again and ONLY write one number such as "3.8".

□ In line 17: the authors mentioned that "Furthermore, the buffer volume was investigated at varied volumes (0.2 – 2.4) and the optimum volume for both approaches was 1.2 ± 0.2. I think as well no need to write the average of the used volume and should be one value, with its metric unit, such as mL. kindly check again.

o In the part of Linearity range and sensitivity:

□ In the sentence "The calibration plot for each methodology", there is a difference between method and methodology, kindly check and select the proper term here.

o In the part Accuracy and precision

□ The authors mentioned that "our concentrations of DLX within linearity range were analyzed and triplicate estimation of each one was recorded. The calculations were listed as % recovery ± SD". Is this study was applied for the standard drug or the extracted sample?? Clarify.

□ In the study of accuracy and precision; the authors used different quality control concentrations, WHY??

□ In table 2 and 3: noted that the value of the higher concentrations was low precise than the low values of concentration. Clarify??

o In the part of Applictiaion

□ The authors mentioned the trade name of the used dosage form of the present drug "Cymbalta". It is better to blind the name of the marketed dosage forms, to avoid the conflict of interest.

□ In the sentence "The estimated percentage values of recovery were 98.02 ± 0.95 and 101.59 ± 1.86 for fluorimetric and RRS approaches". Respectively, 1 mg/capsules (conc/capsules 30 mg) is lost and added, after applied both methods, it is better to mentioned the batch number and manufacture date accordingly.

□ In the part of the "content uniformity test"

• The authors mentioned that "As known from directives of USP, the acceptance value estimated from the following formula $AV = |M - S| + 2.5S$ where; M is a reference value, \bar{X} is mean of recoveries of label claim, S is the SD of recoveries, and K is the constant of acceptability.

- o Kindly put more details and references.
- o What is the value of K is the constant of acceptability, is used during the study.
 - In Table 4: there are values of recoveries (96.17, 96.81, 97.08, ...etc) less than the recorded in the part of the application, using the same cymbatex tablets concentration of, 20 mg/tablet. Clarify the shifted accuracy to these values.
 - In line 22 “cympatex capsules 20 mg tablets”, is correct??
 - The authors mentioned in the part of methods and materials two kinds of the marketed drug “Cymbalta and Cymbatex”, but the content uniformity ONLY applied for one (Cymbatex tablets), could give the reasons.
- Finally, it is better to add further discussion to clarify the obtained results. Other miswriting and suggestion are present throughout the body of the pdf, kindly refer and reply.

Reviewer: 3

Comments to the Author(s)

The manuscript can be accepted after suitable revision. The revised manuscript should address the following comments:

1- Language should be carefully revised throughout the manuscript.

2- Some related articles could be found in the scientific literature. Authors should check these articles, compare their method to those published and cite whatever relevant to their research. Please find them below.

3- Table 1: The intercept values are so large relative to the slope values. I wonder what would be the reason for such high intercept values. Please correct or explain.

Darvishi, E., Shekarbeygi, Z., Yousefinezhad, S., Izadi, Z., Saboury, A.A., Derakhshankhah, H., Varnamkhasi, B.S.

Green synthesis of nanocarbon dots using hydrothermal carbonization of lysine amino acid and its application in detection of duloxetine
(2021) Journal of the Iranian Chemical Society, .

ABSTRACT: Depression is a mood disorder in which a person feels tired and bored and also unwilling to do daily activities. Duloxetine is a drug that is used to the treatment of depression and anxiety. Due to the use of different medications to treat the depression and its possible side effects, quick and accurate identification of these drugs is necessary. Also, because of the possibility of suicide in depressed people, rapid detection of drug type in drug poisoning (drug overdose) is crucial. Therefore, various sensors are used, that the most straightforward, and most accessible sensors are optical types. One of the best, simplest and safest fluorescent sensors were used for optical sensors is nanocarbon dots. In this study, a new, inexpensive and green optical biosensor was designed, and fabricated using lysine-based carbon dots to detect detection of Duloxetine. Fluorescent carbon dot was prepared by hydrothermal method. The green carbon dots were characterized by UV-visible spectroscopy, TEM, XRD and zeta sizer. Also, fluorescence of carbon dot was investigated. The CDs are spherical and the average size of the monodisperse nanoparticles was around 15 nm. The X-ray diffraction pattern represents a weak crystalline property that confirms the amorphous phase of carbon dots. The value of quantum yield for carbon dots was 31.3% to standard Quinone sulfate. The detection limit of Duloxetine was 0.002 μ M. The recovery of Duloxetine was 99.2 to 101.5%, which indicates this nanosensor has a good ability to detect Duloxetine at low concentrations. The results indicate L-lysine-based CDs can be used professionally and selectively to detect of Duloxetine in real samples and human blood plasma. © 2021, Iranian Chemical Society.

Chadha, R., Bali, A.

Stability indicating spectrofluorimetric method for determination of duloxetine hydrochloride in bulk and in dosage form

(2015) *Der Pharmacia Lettre*, 7 (7), pp. 232-240.

ABSTRACT: Duloxetine (DLX), is a selective serotonin-norepinephrine reuptake inhibitor (SNRI) recommended for maintenance treatment of major depressive disorder, neuropathic pain especially diabetic polyneuropathy (first-line treatment), generalized anxiety disorder, stress urinary incontinence and fibromyalgia. The present investigation describes the validation of rapid, sensitive, cost effective and reproducible stability indicating spectrofluorimetric methods based on the native fluorescence of duloxetine HCl in acidic medium for the estimation of duloxetine HCl in bulk and in formulations. The fluorescence intensity of duloxetine hydrochloride was measured at 336 nm after excitation at 290 nm. The methods were validated with respect to linearity, accuracy, precision and robustness. Linearity was observed in the concentration range of 0.3-30 $\mu\text{g/ml}$ with an excellent correlation coefficients (r^2) ranging from 0.9940-0.9996. The limits of assay detection values were found to range from 0.56-0.89 $\mu\text{g/ml}$ and quantitation limits ranged from 1.69-2.42 $\mu\text{g/ml}$ for the proposed methods. The proposed method was applicable to the determination of the drug in capsules and the percentage recovery was found to range from 99.53 \pm 99.66%. The proposed methods were developed as stability indicating procedures by carrying out the analysis for duloxetine hydrochloride on stressed samples prepared under various forced degradation conditions.

Sagirli, O., Toker, S.E., Önal, A.

Development of sensitive spectrofluorimetric and spectrophotometric methods for the determination of duloxetine in capsule and spiked human plasma

(2014) *Luminescence*, 29 (8), pp. 1014-1018.

ABSTRACT: A new, sensitive and selective spectrofluorimetric method has been developed for the determination of duloxetine (DLX) in capsule and spiked human plasma. DLX, as a secondary amine compound, reacts with 7-chloro-4-nitrobenzofurazone (NBD-Cl), a highly sensitive fluorogenic and chromogenic reagent used in many investigations. The method is based on the reaction between the drug and NBD-Cl in borate buffer at pH 8.5 to yield a highly fluorescent derivative that is measured at 523 nm after excitation at 478 nm. The fluorescence intensity was directly proportional to the concentration over the range 50-250 ng/mL. The reaction product was also measured spectrophotometrically. The relation between the absorbance at 478 nm and the concentration is rectilinear over the range 1.0-12.0 $\mu\text{g/mL}$. The methods were successfully applied for the determination of this drug in pharmaceutical dosage form. The spectrofluorimetric method was also successfully applied to the determination of duloxetine in spiked human plasma. The suggested procedures could be used for the determination of DLX in pure form, capsules and human plasma being sensitive, simple and selective. Copyright © 2014 John Wiley & Sons, Ltd.

Alarfaj, N.A., Ammar, R.A., El-Tohamy, M.F.

Cationic-enhanced spectrofluorimetric method for determination of selective serotonin reuptake inhibitor duloxetine hydrochloride in its dosage forms

(2013) *Asian Journal of Chemistry*, 25 (11), pp. [6416-6420](tel:6416-6420).

ABSTRACT: A highly sensitive, rapid, accurate and precise spectrofluorimetric method was developed for the determination of duloxetine hydrochloride in its pharmaceutical formulations. The proposed method is based on investigation of the fluorescence spectral behaviour of duloxetine hydrochloride in cetyl trimethylammonium bromide (CTAB) micellar system. In aqueous solution of borate buffer pH 9.9, the fluorescence intensity of duloxetine hydrochloride was greatly enhanced, 3-fold enhancement, in the presence of cetyl trimethylammonium bromide. The fluorescence intensity of duloxetine hydrochloride was measured at 382 nm after excitation at 275 nm. The fluorescence-concentration plot was rectilinear over the range of 1-70 ng/mL with lower detection limit of 0.5 ng/mL. The method was successfully applied to the analysis of duloxetine hydrochloride in its commercial dosage forms. The results were in good

agreement with those obtained with the reported method. The application of the proposed method was extended to the stability studies of duloxetine hydrochloride after exposure to different forced degradation conditions, such as acidic, alkaline, oxidative and thermal conditions, according to ICH guidelines.

Misiuk, W.

Spectrofluorimetric study on inclusion interaction of β -cyclodextrin with duloxetine and its analytical application

(2012) Indian Journal of Chemistry - Section A Inorganic, Physical, Theoretical and Analytical Chemistry, 51 (12), pp. 1706-1710.

ABSTRACT: Fluorescence study on inclusion interaction of duloxetine in β -cyclodextrin shows significant increase in the fluorescence of duloxetine in the presence of β -cyclodextrin. The effects of pH and cyclodextrin concentration on the fluorescence spectra are reported.

Fluorescence spectroscopy of the host-guest interaction between duloxetine and β -cyclodextrin shows formation of inclusion complex with a 1:1 stoichiometric ratio. The changes in fluorescence of duloxetine on inclusion in the hydrophobic β -cyclodextrin cavity is used to calculate its association constants by non-linear regression method. A new spectrofluorometric method is proposed for quantitative determination of duloxetine in dosage forms with limit of detection of 2.0×10^{-8} mol/L. The linear range is 5.16×10^{-8} mol/L to 1×10^{-5} mol/L and correlation coefficient is 0.9998. The accuracy (recovery 99.5-100.8 %) and precision (RSD 0.58-0.89 %) values of the proposed method are satisfactory.

Ulu, S.T.

Determination and validation of duloxetine hydrochloride in capsules by HPLC with pre-column derivatization and fluorescence detection

(2012) Journal of Chromatographic Science, 50 (6), pp. 494-498.

ABSTRACT: A high-performance liquid chromatographic (HPLC) method is described for the determination of duloxetine hydrochloride in capsules. The method was based on pre-column derivatization with 4-chloro-7-nitrobenzo-2-oxa-1,3-diazole using the fluorimetric detection technique. Duloxetine hydrochloride was analyzed by HPLC using an Inertsil C18 column (5 μ m, 150 \times 4.6 mm) and mobile phase consisted of methanol and water (65:35, v/v). The fluorescence detector was adjusted at excitation and emission wavelengths of 461 and 521 nm, respectively. The linearity of the method was in the range of 10-600 ng/mL. Limits of detection and quantification were 0.51 and 1.53 ng/mL, respectively. The proposed method was successfully applied for determination of duloxetine hydrochloride in its pharmaceutical preparation. The results were in good agreement with those obtained using a reference method. © The Author [2012].

Liu, X., Du, Y., Wu, X.

Study on fluorescence characteristics of duloxetine hydrochloride

(2008) Spectrochimica Acta - Part A: Molecular and Biomolecular Spectroscopy, 71 (3), pp. 915-920.

ABSTRACT: The fluorescence characteristics of duloxetine hydrochloride are studied in this paper. The fluorescence emission spectra of duloxetine demonstrate that intramolecular charge-transfer takes place between thiophene ring and naphthalenyloxy group upon irradiation. The effects of excitation light, solvent system, variation of solution pH value, metal ions and vitamin C on the fluorescence spectra of duloxetine hydrochloride are elucidated, respectively. A spectrofluorometric method of quantitative determination of duloxetine in dosage form is reported for the first time, the linear range is 7.14×10^{-8} mol/L to 1.43×10^{-5} mol/L, the linear correlation coefficient r is equal to 0.9997, and the detection limit is 3.5×10^{-8} mol/L. The accuracy and the precision are satisfactory. © 2008 Elsevier B.V. All rights reserved.

Prabhu, S., Shahnawaz, S., Kumar, C., Shirwaikar, A.

Spectrofluorimetric method for determination of duloxetine hydrochloride in bulk and pharmaceutical dosage forms

(2008) Indian Journal of Pharmaceutical Sciences, 70 (4), pp. 502-503.

ABSTRACT: A simple accurate, sensitive and reproducible spectrofluorimetric method was developed for the analysis of duloxetine hydrochloride in pure and pharmaceutical dosage form. Duloxetine hydrochloride showed strong native fluorescence in 0.05 M acetic acid having excitation at 225 nm and emission at 340 nm. Effect of different solvents were thoroughly investigated. The calibration graph was linear in the range from 0.020 to 0.400 μ g/ml. The proposed method was statistically validated and successfully applied for analysis of capsule dosage forms. The limit of detection and limit of quantification were found to be 0.003 μ g/ml and 0.010 μ g/ml, respectively. The percentage recovery was found to be in the range of 98.71% to 99.17%.

Author's Response to Decision Letter for (RSOS-210922.R0)

See Appendix A.

RSOS-210922.R1 (Revision)

Review form: Reviewer 1 (Walaa Abd-ALGhafar)

Is the manuscript scientifically sound in its present form?

Yes

Are the interpretations and conclusions justified by the results?

Yes

Is the language acceptable?

Yes

Do you have any ethical concerns with this paper?

No

Have you any concerns about statistical analyses in this paper?

No

Recommendation?

Accept as is

Comments to the Author(s)

This is a clear, concise, and well-written manuscript. It is useful in quality control laboratories.

Review form: Reviewer 2

Is the manuscript scientifically sound in its present form?

Yes

Are the interpretations and conclusions justified by the results?

Yes

Is the language acceptable?

Yes

Do you have any ethical concerns with this paper?

No

Have you any concerns about statistical analyses in this paper?

No

Recommendation?

Accept as is

Comments to the Author(s)

I don't have any comments again, the authors reply my pervious comments perfectly

Review form: Reviewer 3

Is the manuscript scientifically sound in its present form?

Yes

Are the interpretations and conclusions justified by the results?

Yes

Is the language acceptable?

Yes

Do you have any ethical concerns with this paper?

No

Have you any concerns about statistical analyses in this paper?

No

Recommendation?

Accept as is

Comments to the Author(s)

No more comments. Manuscript accepted.

Decision letter (RSOS-210922.R1)

Dear Mr Abu-hassan:

Title: Two facile approaches based on association complex with erythrosine-B for nano-level analysis of duloxetine. Application to content uniformity
Manuscript ID: RSOS-210922.R1

It is a pleasure to accept your manuscript in its current form for publication in Royal Society Open Science. The chemistry content of Royal Society Open Science is published in collaboration with the Royal Society of Chemistry.

Yours sincerely,
Dr Ellis Wilde
Publishing Editor, Journals

RSC Associate Editor
Comments to the Author:
(There are no comments.)

RSC Subject Editor
Comments to the Author:
(There are no comments.)

Reviewer(s)' Comments to Author:

Reviewer: 3

Comments to the Author(s)

No more comments. Manuscript accepted.

Reviewer: 2

Comments to the Author(s)

I don't have any comments again, the authors reply my pervious comments perfectly

Reviewer: 1

Comments to the Author(s)

This is a clear, concise, and well-written manuscript. It is useful in quality control laboratories.

Appendix A

Responses to reviewer's comments

Dear Prof Dr. Editor of **Royal Society Open Science**.

On the behalf of all authors, I would like to thank you for the opportunity that we have been given to further revise our manuscript.

Manuscript ID: RSOS-210922

Title "**Two facile approaches based on association complex with erythrosine-B for nano-level analysis of duloxetine. Application to content uniformity**" for publication in Journal: **Royal Society Open Science**.

The manuscript was carefully revised according to the comments of the reviewers. We are grateful for the comments and advice we have received. Please find enclosed the revised version of our manuscript. As seen below each reviewer's comment(s) will be mentioned followed by our response.

N. B. All corrections have been re-written in red color in the revised manuscript.

Reviewer: 1

-Who is the corresponding author? Please refer to him in affiliation.

Done

- The word (xanthene) in the paragraph under Results and discussion should be (Xanthene) first letter is capital.

Done

- There should be 10 spaces at the beginning of the paragraphs in all manuscript.

Done (N.B The paper will be rewritten and edited according to journal format by the publisher).

Reviewer 2

- **In the part of the Abstract:**

o Line 29: Kindly add the abbreviation of Resonance Rayleigh Scattering, RRS.

done

o In the abstract, line 34, the linearity was 100 – 2400 ng/mL, it is better to consider the number significance writing to be replaced by 0.1 – 2.4 µg/mL. In the same context, the linearities are nearly the same as 0.1 or 0.2 to 2.4 or 2.0 µg/mL, for both methods spectrofluorimetry and RRS, respectively. BUT it was expected that if you reached the ideal conditions of the resonance light-scattering (RLS) technique, the optical properties of the resonance light scattering has a magnitude higher than light emission, could you explain that WHY?

The unit has been changed to µg/mL. the linear ranges were derived after careful optimization of the conditions and although RRS has a higher magnitude the fluorometric method exhibit higher sensitivity due to the value of blank which subtracted from all results value.

o The LOD of the fluorimetric method is 0.030 µg/mL, BUT for RRS was 0.056 µg/mL, that the fluorimetric method is more sensitive, why extend the other procedure, RRS, give the resonance.

The LOD calculation is dependant on the value of standard deviation of the intercept and slope according to ICH guidelines which after calculation exhibit that the fluorimetric method is more sensitive than RRS.

- **In the part of the Introduction**

o In line 19: replace the word “methodology” with “method”, kindly, as you talked about the surveys, interviews, experiments, and other investigatory processes, etc.

Done

- **In the part of Materials, standard solution, and reagents_line 27**

o The mention names of the dosage forms, Cymbalta and Cymbatex, it is advised to remove the trad name to guard your work against any conflict of interest, related to these companies

Please see all reported methods for DLX analysis you will find that trade name is mentioned in materials and experimental.

o In writing of the standard solution, you have ONLY to write the final concentrations of stock solutions, and working solution, or quality control sample as well. Kindly remove the un-wanted details.

Done. Please see the revised manuscript.

- **In the part of general assay procedure:**

o The used indicative concentration units are different, for drug ng/mL, and for the reagent was molarity, it is better to unify these units.

Done, please see the revised manuscript.

- **In the the Results and discussion part:**

o Line 39, use the name erythrosine-B in the whole manuscript.

Done.

- **In the part of Results and Discussion**

o **In line 9:** you did not mention the fixed value of $\Delta\lambda$, nm, is used in synchronous activity??

The value of $\Delta\lambda$ has been mentioned in the part of the experimental.

- **In the part of Mechanism of augmentation of resonance Raleigh scattering**

o **In the equation of**

▪ $I = KCM I_0$ where I represent RRS strength, K coefficient constant, M M.Wt, I the incident intensity, C refer to solution concentration.

The sentence has been rephrased and edited.

▪ HOW you calculated all these variables??? You have to clarify more discussion.

The equation aims to confirm that the formation of ion-pair leads to an increase in molecular weight which in turn leads to an increase in the RRS signal as other variables is constant.

- M.Wt: is this an official abbreviation??

The abbreviation has been removed.

o In line 38: “when Erythrosine reacts with DLX via association complex, a hydrophobic solid-liquid interface appeared leads to an augmentation of the RRS signal”?? clarified the formed complex in the form of nanoparticles of discussion that is required for RRS?? Mention more discussion??

The Ion-pair complex is a well-known reaction formed by electrostatic attraction between oppositely charged ions by electrostatic attraction. The formed complex will neutralize ions and so increase hydrophobicity and solid-liquid interface.

o **In line 48:** “consequence, the enhancement of the scattering intensity was achieved”. In the reported reference you cited, they studied the interaction of erythrocin-B with R- and S-propranolol; the drug was used is different than DLX, did you confirm the rigidity of the formed complex by another way,?? Clarify??

Formation of ion-pair complex leads to increase of molecular weight of the compound and so the rigidity will be increased. Please see the following articles[1-3]

- **In the part of Experimental conditions study:**

o **In the part of the Buffre pH and its volume:**

▪ **In line 15:** the written pH value was “pH 3.8 ± 0.2 ”, I think NO mean to calculate the pH average?? Kindly check the value again and ONLY write one number such as “3.8”.

See the revaised manuscript.

▪ In line 17: the authors mentioned that “Furthermore, the buffer volume was investigated at varied volumes (0.2 – 2.4) and the optimum volume for both approaches was 1.2 ± 0.2 . I think as well no need to write the average of the used volume and should be one value, with its metric uint, such as mL. kindly check again.

Done.

o **In the part of Linearity range and sensitivity:**

- In the sentence “The calibration plot for each methodology”, there is a difference between method and methodology, kindly check and select the proper term here.

Done

o **In the part Accuracy and precision**

- The authors mentioned that “our concentrations of DLX within linearity range were analyzed and triplicate estimation of each one was recorded. The calculations were listed as % recovery \pm SD”. Is this study was applied for the standard drug or the extracted sample?? Clarify.

The study was applied for standard drug as known from ICH guidelines. See the revised manuscript.

- In the study of accuracy and precision; the authors used different quality control concentrations, WHY??

The guidelines of ICH state that accuracy and precision should be investigated at least at three concentration levels using three determinations for each.

- In table 2 and 3: noted that the value of the higher concentrations was low precise than the low values of concentration. Clarify??

All values in the table are within the accepted range at the confidence limit 95 % and variation is due to normal fluctuation of the result.

o **In the part of Application**

- The authors mentioned the trade name of the used dosage form of the present drug “Cymbalta”. It is better to blind the name of the marketed dosage forms, to avoid the conflict of interest.

Kindly, see all reported methods for determination of the drug which mentioned the trade name of dosage form.

- In the sentence “The estimated percentage values of recovery were 98.02 ± 0.95 and 101.59 ± 1.86 for fluorimetric and RRS approaches”. Respectively, 1 mg/capsules (conc/capsules 30 mg) is

lost and added, after applied both methods, it is better to mentioned the batch number and manufacture date accordingly.

Done.

• **In the part of the “content uniformity test”**

• The authors mentioned that “As known from directives of USP, the acceptance value estimated from the following formula $AV = |M - X| + KS$ where; M is a reference value, X is mean of recoveries of label claim, S is the SD of recoveries, and **K is the constant of acceptability.**

o Kindly put more details and references.

Done

o What is the value of K is the constant of acceptability, is used during the study.

Please see the revised manuscript

• **In Table 4:** there are values of recoveries (96.17, 96.81, 97.08, ...etc) less than the recorded in the part of the application, using the same cymbatex tablets concentration of, 20 mg/tablet. Clarify the shifted accuracy to these values.

This is due to the normal fluctuation of the result and variation of the formulation process. Moreover, the calculation of AV indicates that all result is acceptable.

• In line 22 “cympatex capsules 20 mg tablets”, is correct??

yes

• The authors mentioned in the part of methods and materials two kinds of the marketed drug “Cymbalta and Cymbatex”, but the content uniformity ONLY applied for one (Cymbatex tablets), could give the reasons.

According to USP guidelines, the uniformity of the drug should be tested when the concentration is smaller than 25 mg so Cymbatex 20 mg only was chosen for the CU study.

- Finally, it is better to add further discussion to clarify the obtained results. Other miswriting and suggestion are present throughout the body of the pdf, kindly refer and reply.

The manuscript is deeply revised and edited. please see the revised manuscript.

Reviewer 3

The manuscript can be accepted after suitable revision. The revised manuscript should address the following comments:

1- Language should be carefully revised throughout the manuscript.

The manuscript is deeply revised and edited.

2- Some related articles could be found in the scientific literature. Authors should check these articles, compare their method to those published and cite whatever relevant to their research. Please find them below.

The mentioned references have been cited in the manuscript and in a tabular form, a comparison of the proposed method with the mentioned methods was performed.

3- Table 1: The intercept values are so large relative to the slope values. I wonder what would be the reason for such high intercept values. Please correct or explain.

The higher value of the intercept is due to the high value of blank. In addition, the reading of the utilized instrument is relatively higher than others as it is up to 60,000. The low value of the slope because the utilized unit is a nanogram when the unit changed to microgram as recommended from reviewer 2 the value of slope will be 570 and 1762 respectively. Please see table 1 in the revised version.

[1] M.A. Abdel-Lateef, S.M. Derayea, D.A.N. El-Deen, A. Almahri, M.J.R.S.o.s. Oraby, Investigating the interaction of terbinafine with xanthenes dye for its feasible determination applying the resonance Rayleigh scattering technique, 8 (2021) 201545.

[2] M.A. Abdel-Lateef, A. Almahri, S.M. Derayea, E.J.R.i.A.C. Samir, Xanthene based resonance Rayleigh scattering and spectrofluorimetric probes for the determination of cyclobenzaprine: application to content uniformity test, 39 (2020) 222-230.

[3] A. Almahri, M.A. Abdel-Lateef, E. Samir, S.M. Derayea, M.A.J.L. El Hamd, Resonance Rayleigh scattering and spectrofluorimetric approaches for the selective determination of rupatadine using erythrosin B as a probe: application to content uniformity test, 36 (2021) 651-657.